# CDR1 Composition Can Affect Nanobody Recombinant Expression Yields

**DOI:** 10.3390/biom11091362

**Published:** 2021-09-14

**Authors:** Marco Orlando, Sara Fortuna, Sandra Oloketuyi, Gregor Bajc, Adi Goldenzweig, Ario de Marco

**Affiliations:** 1Department of Biotechnology and Life Sciences, University of Insubria, Via J. H. Dunant 3, 21100 Varese, Italy; m.orlando14@campus.unimib.it; 2Department of Chemical and Pharmaceutical Sciences, University of Trieste, Via L. Giorgieri 1, 34127 Trieste, Italy; s.fortuna@units.it; 3Lab of Environmental and Life Sciences, University of Nova Gorica, Vipavska cesta 13, Rožna Dolina, 5000 Nova Gorica, Slovenia; sandra.folarin.oloketuyi@ung.si; 4Department of Biology, Biotechnical Faculty, University of Ljubljana, Večna pot 111, 1000 Ljubljana, Slovenia; gregor.bajc@bf.uni-lj.si; 5Department of Biomolecular Sciences, Weizmann Institute of Science, Rehovot 7610001, Israel; adi.goldenzweig@weizmann.ac.il

**Keywords:** nanobody CDRs, in silico modeling, rational mutagenesis, nanobody engineering

## Abstract

The isolation of nanobodies from pre-immune libraries by means of biopanning is a straightforward process. Nevertheless, the recovered candidates often require optimization to improve some of their biophysical characteristics. In principle, CDRs are not mutated because they are likely to be part of the antibody paratope, but in this work, we describe a mutagenesis strategy that specifically addresses CDR1. Its sequence was identified as an instability hot spot by the PROSS program, and the available structural information indicated that four CDR1 residues bound directly to the antigen. We therefore modified the loop flexibility with the addition of an extra glycine rather than by mutating single amino acids. This approach significantly increased the nanobody yields but traded-off with moderate affinity loss. Accurate modeling coupled with atomistic molecular dynamics simulations enabled the modifications induced by the glycine insertion and the rationale behind the engineering design to be described in detail.

## 1. Introduction

The increasing interest in single-domain antibodies originated from the discovery of the *Camelidae* IgG2 and IgG3 VHH domains (nanobodies) and was due to their structural properties [1] as well as to their advantages for recombinant expression. The latter enables the design of large pre-immune libraries containing clones with favorable biophysical characteristics and that are fused to suitable tags [2,3,4,5], the elaboration of in vitro selection protocols for the identification of clones possessing particular features [6], and the simple post-selection engineering of the promising candidates into application-optimized reagents [7]. What still represents a major bottleneck in the nanobody selection process is the often-observed stability discrepancy between the binders secreted on the surface of phages and the same clones expressed recombinantly in bacteria. This bottleneck often leads to the loss of several potentially interesting binders during the post-panning steps. Although the empiric optimization of expression yielded by testing combinations of strains, expression vectors and growth conditions are feasible, but these processes are extremely laborious or limited to those few laboratories dedicated to protein expression and that are supported by automatized high throughput platforms [8]. The possibility of improving clone stability, increasing their yields, and maintaining or improving the sequence humanization by rational mutagenesis would represent a substantial advancement, and community efforts have already resulted in the development of successful approaches. There are several examples demonstrating the feasibility of this approach, but these are still the results of case-by-case projects that require a major input from specialized researchers and often require access to structural information [9,10,11,12]. On the other hand, we recently confirmed that publicly accessible algorithms that require minimal data inputs can rapidly and effectively help in the identification of protein variants that exhibit astonishingly higher yields than the original sequences [13]. A major challenge for stability design methods is the preservation of the protein function despite the introduction of stabilizing mutations. This issue is particularly critical for nanobodies since their paratopes involve a relatively large portion of the complete protein, and the amino acids participating directly or indirectly in the antigen binding are distributed over several, non-continuous sequences of both complementary determining regions (CDRs) and frameworks [14,15].

PROSS (protein repair one-stop shop) is an automated structure- and sequence-based design method for optimizing protein stability and heterologous expression levels [16]. Once the data of the target protein are inserted, PROSS provides a list of sequences that contain an increasing number of point mutations. To reduce the risk of disrupting the protein’s biological function, the user is recommended to specify active-site regions or residues. We previously used the anti-Her2 ectodomain nanobody A10 [4,17] with affinity in the low nM range as a model to evaluate in silico optimization approaches [12] and therefore also considered it to be the most suitable candidate to undergo a PROSS-mediated mutagenesis strategy. At the time at which this project started, no structural model was yet available to describe the interaction between A10 and the Her2, and it was assumed that the nanobody paratope was restricted to the amino acids in the CDRs. Therefore, we disallowed mutagenesis at all CDR positions. Technically, this is simple to obtain because the PROSS interface enables the specification of selected amino acids that the algorithm will not propose mutations for. Under these conditions, the program proposed a set of point mutations in the nanobody framework, and the corresponding variants were expressed and evaluated, and the results were discussed in a recent paper [13]. Here, we further assessed the capacity of the PROSS to suggest fitter mutants by removing the protection for the residues present in the CDRs. This condition enabled the identification of a critical solubility spot in the CDR1 and was instrumental to start the analysis of the structural characteristics of this loop and to conceive a strategy for producing a variant that optimizes the trade-off between affinity and stability.

## 2. Materials and Methods

### 2.1. PROSS-Based Nanobody Variant Identification and Production

The PROSS method [16] is publicly available at https://pross.weizmann.ac.il (accessed on 28 June 2018) and requires a structure or an accurate 3D model as input. A 3D model of A10 was generated by homology modelling using chain B PDB ID 6ey0, which shares ≈ 70% sequence identity with A10. Overall, two submissions of the A10 model were made to the PROSS server. The first only enabled mutations at framework residues (positions 1–29, 36–54, 62–100 and 110–122, described in [13]), and the third submission allowed the design of the complete nanobody, including the CDR amino acids. The results of the latter served as a reference for selecting the variants to experimentally express and analyze (Table 1). The sequences corresponding to A10 mutKG.1 and A10 mutKG.2 were synthetized by Doulix (Venezia, Italy), whereas A10 mutKG and A10 mutG0 were synthetized by Twist Bioscience (San Francisco, CA, USA). All of the constructs were subcloned into pET-derived vectors and were produced as previously reported [17].

### 2.2. Surface Plasmon Resonance (SPR) Experiments

The VHH binding affinity was measured with a Biacore T200 (GE Healthcare, Uppsala, Sweden). The temperature was kept at 25 °C, and the data were fitted with a 1:1 Langmuir interaction model. The ligand (Her2 ectodomain-Fc) was diluted to 50 µg/mL in sodium acetate buffer, pH 5.0, before being immobilized by amine-coupling on a CM5 chip (GE Healthcare, Uppsala, Sweden) at 1180 RU. Nanobodies were diluted in 20 mM Hepes, 150 mM NaCl, 3 mM EDTA, and 0.005% Tween 20 at pH 7.4 before injection at 30 µL/min. Eight concentrations between 250 and 3.5 nM were tested, and kinetics were collected in a unique sequence of injections. Data were analyzed by means of the Biacore T200 Evaluation Software (version 3.2).

### 2.3. SeqDesign Analysis of Nanobody Sequences)

The log-ratio of individual sequence likelihoods ($\log\frac{p(x^{Mutant}|\theta )}{p(x^{WT}|\theta )},\;\mathrm{called}\;\mathrm{relative}\;\mathrm{sequence}\;\mathrm{fitness}$) has been previously used to predict mutation effects such as expression yield and stability variations starting from the probability (p) of mutant ($x^{Mutant}$) and *WT* ($x^{WT}$) sequences, given a parameterized (*θ*) model [18,19].

SeqDesign employs word embedding and an autoregressive feed-forward deep generative neural network model for predicting specific amino acid probability at any position in the sequence given the previous amino acids present in the sequence. Thus, full sequence probability is obtained by the product of conditional probabilities on previous characters along a sequence (see Equation (1) in [20]). This allows probability predictions on insertional variants to be dealt with. Three models were trained on the nanobody sequence dataset for 1000 iterations to predict the probabilities of A10 variants and *WT* and to estimate the relative sequence fitness (training and predicting scripts available at https://github.com/debbiemarkslab/SeqDesign; accessed on 1 December 2019).

### 2.4. PEP-FOLD3-Based Analysis

The sequence GSLRLSCAAS(G)ATSGNISNMGWFRQA, which corresponds to the G18-A42 (G18-A43) region of A10 (A10 mutG0) and includes the CDR1 loop comprising aa 27–33 (27–34), were submitted to the PEP-FOLD3 server [21]. The output was analyzed without further optimization.

### 2.5. Molecular Dynamics Simulations (Free Nanobodies)

The A10 starting model was prepared by homology followed by minimization and short atomistic molecular dynamics (MD) equilibrations in full water solvent as in [22]. Na^+^ and Cl^−^ were employed as counterions to neutralize the system. A10 mutG0 was constructed by homology employing the equilibrated A10 model as a template. The above process was repeated for this mutant. The equilibrated systems, 26,289 atoms for A10 and 26,255 for A10 mutG, underwent 1000 ns NPT production runs at 300 K. The time step was set to 2 fs with the Verlet integrator and LINCS [23] constraint. The particle mesh Ewald summation accounted for long range electrostatic interactions. The temperature was controlled with a modified Berendsen thermostat [24], and the pressure had an isotropic Parrinello-Rahman of 1 bar. An AMBER99SB-ILDN [22] force field and tip3p water were employed. Configurations and energies were sampled every 1.0 ns. All of the simulations and their analysis were run as implemented in the GROMACS package [25].

### 2.6. Prediction of Binding Free Energy Change upon Mutation

The A10 wt/Her2 starting model, also employed as a template for the mutants, was reported in [15]. Complexes with MutKG, mutG0, and mutKG.1 were obtained by mutating the above template with Modeller 9.25 [26]. The loop refinement protocol was used on the mutated CDR1 loop to add the glycine insertion. Ten models were generated, and the one with the best DOPE score was selected. All of the models were energy minimized in a vacuum by employing the steepest descent algorithm as implemented in the GROMACS package [23]. CHARMM36 force field (February 2021 version, downloaded from http://mackerell.umaryland.edu/charmm_ff.shtml#gromacs [27] accessed on 15 March 2021) was employed. The protonation states were determined by PDB2PQR [28], estimating the titration states by propka 3.1 (https://github.com/schrodinger/propka-3.1 accessed on 15 March 2021) under the same force field at pH 7.4 [29,30]. The hydrogenated complex was solvated in a dodecahedron box containing TIP3P-parameterized explicit water molecules and Na^+^ and Cl^−^ ions at a concentration of 150 mM to neutralize the system charges. Five independent MD simulations were performed for each system by employing the same protocol, including energy minimization and the position-restrained equilibration of the solvated system prior to running each unbiased simulation, as outlined by Lindahl for lysozymes in water [31]. Each system was equilibrated under NPT conditions at 25 °C (V-rescale thermostat) and 1 atm (Berendsen barostat) by scaling all of the coordinates with the scaling matrix of the pressure coupling. Two ns of MD were performed without position restraints to Her2 and by reducing the position restraints on A10 from 1000 to 150 kJ mol^−1^ nm^−1^. The same protocol was applied for another 2 ns, inverting the protein restraints, namely restraining only Her2 and releasing A10, before removing any restraint and performing an unbiased simulation for 20 ns with an integration step of 2 fs. Covalent bonds between hydrogens and heavy atoms were constrained by the LINCS algorithm. The information on the system was saved every 0.2 ns. Major motions were investigated by cluster analysis performed with the gromos method (cut-off 2 Å) on the CDR1 atoms of backbone-superposed A10 simulation frames.

The estimated binding free energy (ΔG) for the Her2/A10 complex formation was calculated using MMPBSA.py from the Amber19 package [32] by employing the MM-GBSA method [33]. The single-trajectory approach was used, exploiting trajectories for the free proteins extracted from the evaluation of the complex. A generalized born implicit-solvation model (GBOBC2) was employed [34].

ΔG value was estimated over each independent trajectory together with its standard deviation. Per-residue binding contributions were calculated by the per-residue effective free-energy decomposition (prEFED) protocol implemented in MMPBSA.py [30], ignoring the entropic contribution. Per-residue binding free energy change (∆∆G_“mut”) was calculated as ∆∆G_mut_ = 〈∆G_mut_〉 − 〈∆G_WT_〉, where 〈∆G_x_〉 were averages calculated over replicas. Positive (or negative) values indicate a decrease (increase) of the predicted interaction energy. PyContact [35] was used to detect non-covalent interactions between the Her2 and A10 variants along MD frames sampled every 0.2 ns with a distance threshold of 4 Å. ChimeraX 0.93 (https://www.rbvi.ucsf.edu/chimerax/ accessed on 8 August 2021) was used to prepare the figures [36,37].

## 3. Results and Discussion

A10 is an anti-Her2 nanobody isolated by panning a synthetic library in which the clones only differ in the CDR positions [4] and was previously selected as a model for benchmarking the performance of the PROSS server [13]. While only mutations in the framework were assessed to preserve the CDR-based paratope in the benchmark work, here, we employed PROSS to identify fitter mutants by considering the whole VHH sequence. Surprisingly, under these conditions, PROSS proposed a variant with five mutations, all concentrated in the CDR1, which comprises only 7 aa (Figure 1a: from ATSNISN to RTFSIYD). This outcome implies that CDR1 is likely to be an instability hot spot. On the other hand, its interaction with Her2 is known to have a stabilizing effect [14]. Indeed, this is one of the main regions involved in the interaction between A10 and Her2, as identified by combining NMR and cross-linking experiments with a modeling approach [14]. Specifically, it was confirmed that the CDR1 residues A29, T30, I33, and S34 contributed to the nanobody paratope. We hypothesized that massively mutating CDR1 would abrogate binding to the antigen. Instead, we aimed at more moderate modifications to CDR1 that may improve the nanobody stability while preserving its binding capacity.

We started looking for working hypotheses by analyzing the structural properties of CDR1 in isolation from the nanobody scaffold. More precisely, the sequence G18-A42 (Figure 1b) comprising the A10 wt CDR1 (residues A29-N35) together with the beta strands preceding and following it was evaluated using PEP-FOLD3, an algorithm developed to analyze the dynamic folding of 5 to 50 residue long peptides. PEP-FOLD3 predicted that the peptide assumed a set of structured hair-pin-like conformations that might represent a potential element of instability (Figure 1c). This assumption derives from the consideration of the peculiar characteristics of synthetic nanobodies such as A10, in which the randomly generated sequences corresponding to CDRs are forced inside fixed scaffolds. Such CDR/scaffold combinations underwent no evolutionary/functional optimization; therefore, they must be able at least to adapt to the surrounding framework residues [14]. Consequently, despite the fact that structured domains are more stable than flexible regions in general, in the case of synthetic nanobodies, more relaxed states can provide more stability options than strained conformations. We reasoned that, instead of mutating the CDR1 residues, with the risk of strongly reducing the nanobody affinity for its antigen, we could improve its structural stability by providing more flexibility to the loop by inserting a glycine along its sequence. Glycine was chosen because of its small dimension and because it enables the rotation of the polypeptide chain. Indeed, structural modeling showed that the addition of glycine at various positions across CDR1 resulted in a corresponding “relaxed” conformation (Figure 1d and Appendix A for the complete list of the mutants). We thus evaluated the effect of such variants by running PROSS on them. Interestingly, the PROSS output for all of the input variants (regardless of the extra-glycine position) did not include mutations in CDR1 or its vicinity suggesting that, in contrast to A10 wt, the new variants are stable. We then evaluated the effect of glycine insertion on the selected CDR1 positions by the recently developed method SeqDesign [20], which predicted a higher relative sequence fitness for all of the possible glycine insertion combinations with respect to the A10 wt taken as a reference (its fitness was set to 0, Appendix A).

The conformational modifications induced in the CDR1 were assessed by analyzing the mutant A10 mutG0 in detail, as this insertion was expected to minimally affect the binding to the antigen. Indeed, this mutant presents the addition of a glycine after S31, namely intermediately between residues 29–30 and 33 which, according to NMR data [15], are directly participating with the paratope (Figure 1d). Specifically, we analyzed the portion of the sequence containing the insertion, namely the CDR1 and part of the up- and downstream confining framework residues that act as fixing points for the CDR1 loop and evaluated its flexibility. We assumed that a larger conformational freedom could correlate with a larger yield, as a peptide capable of exploring multiple conformations can find its optimum fold faster. The sequence G18-A43 was analyzed through the PEP-FOLD3 server (Figure 1e). In contrast to the results obtained using the A10 wt homologous sequence that indicated the presence of structured hair-pin-like conformations (Figure 1c), the corresponding A10 mutG0-derived conformations remained unstructured and were able to explore a large diversity of structural options (Figure 1e). This evidence of higher flexibility and capacity to rearrange its own structure may explain why mutG0 would provide higher stability with respect to the relatively stiff CDR1 of A10 wt. To confirm this statement, we first further explored how the higher flexibility of the G-containing mutant affected the VHH folding by running 1 μs long atomistic molecular dynamic simulations of each nanobody in water solvent and analyzed the obtained trajectories (Figure 2). Along the simulated time, both A10 wt and A10 mutG0 explored a number of conformations, as evidenced by their backbone root mean squared deviation (RMSD, Figure 2a,b), which gives an indication of how much protein backbone atoms collectively displace from their initial configuration along the simulated time. Interestingly, the wild type protein emerges as a very dynamic system, and its backbone appeared more flexible than the G-mutated one: its RMSD reaches values up to 3 Å, while that of A10 mutG0 settled at about 2 Å. This flexibility did not affect the overall protein shape, as the backbone radius of the gyration remained constant (13.5 Å) in both proteins (Figure 2c,d). Their backbone root mean squared fluctuation (RMSF, Figure 2e,f) allowed the fact that the residues that underwent major backbone rearrangements were in the CDRs (RMSF = 0.45 Å) to be pinpointed. Interestingly, CDR1 and CDR2 were more mobile in A10 mutG0 than in A10 wt, while the opposite was observed for CDR3. This can be appreciated by looking at the VHHs conformations along the trajectories (Figure 2g,h) and can be measured by the CDRs RMSD (Figure 2i–k for A10 wt and Figure 2l–n for A10 mutG0). As suggested above by the peptide analysis (Figure 1), the addition of a glycine at position 32 led to a longer, more mobile, CDR1 that, in return, affected the rearrangement of the other CDRs: the short CDR2 bent to leave space for the elongated CDR1, while CDR3 become sterically arrested. This resulted in a different binding surface. Aware of the limits of this analysis, resulting from the data of a single (even if reasonably long) MD run, we believe the enhanced mobility observed in the CDR might explain the five-fold decrease in binding affinity to the target later observed in the experiments.

Next, we moved to the experimental validation of the selected mutants (Table 1). We explored two representative sequences: A10 mutG0 and another one in which the glycine insertion was just after the first residue (A29) of the CDR1 (A10 mutKG) because it had the best SeqDesign score (7.66 ± 0.16). Furthermore, two A10 mutKG variants were considered. The clone A10 mutKG.1 had five additional point mutations in its framework, originally suggested by PROSS for A10 wt (in the submission that allowed design at the framework alone) and corresponding to the most represented residues in camelid heavy-chain single-domains (sequence “camelization”). The clone A10 mutKG.2 had the same five mutations plus another mutation proposed by PROSS in CDR3 (Table 1). SeqDesign analysis could not be applied to these last two mutants because it has not been conceived to evaluate multiple mutations in conserved regions.

The A10 mutG0 clone (6.9 mg/L medium) produced roughly 30% more than A10 wt (5.2 mg/L), but both A10 mutKG and A10 mutKG.1 had higher production yields (9.6 and 12.8 mg/L, respectively). Interestingly, whereas the mutations from “human-like” to the canonical “llama” residues in the framework of A10 wt did not result in higher production yields [12], the mutant with glycine insertion (A10 mutKG.1) profited from the “camelization” of the original partially humanized sequence (A10 mutKG). In contrast, the further mutation present in A10 mutKG.2 (K/L in CDR3) had a remarkable negative effect on the nanobody yield, suggesting that the wt CDR3 was already stable and did not require further optimization. This clone was consequently dismissed. The linear regression data (R2 = 0.90) correlating to the expression yield (mg/mL) of the A10 mutants and their corresponding sequence fitness (Appendix A) indicated the capacity of SeqDesign to predict the mutation effect on the clone solubility.

To evaluate how the glycine insertion position could affect the nanobody binding capacity to its antigen, mutKG, mutG0, and mutKG.1 were docked to Her2 using the structure of the A10 wt/Her2 complex predicted by NMR/crosslink mass spectrometry and refined by MD as a reference [15] (Appendix A). Her2 structural models with a higher resolution (3MZW and 3BE1) could not be considered for this study because they do not comprise the Her2 domain predicted to interact with A10, whereas 6J71 was not available at the time in which this work was performed. However, despite the fact that the 6OGE resolution is lower than that of 6J71, it was refined and deposited at atom-level resolution and not as a protein outline. Moreover, the docking/MD results of the Her2/A10 complex obtained by using this model were compatible with experimental data [15], indicating its reliability.

The models underwent a set of MD simulations from which the mutation-dependent ΔΔG in the CDR1 region was estimated (Figure 3a). Calculated results indicated that glycine insertion between A29 and T30 (mutKG and mutKG.1) are expected to reduce the relative contribution of A29 to the paratope (predicted ΔΔG ≈ 0.87/1.02 ± 0.39/0.63), and the loss is compensated by the concomitant binding strength increase of T30 to Her2 (ΔΔG ≈ −1.35/−1.24 ± 0.13/0.15). A29 is displaced from its contact with Her2 C563-C576 disulfide bridge, whereas T30 acquires a stacking hydrophobic interaction with F595 (Figure 3b). Furthermore, the A10-Her2 model [15] suggests that a glycine inserted between A29 and T30 should be exposed to and interact with the Her2 Y554 ring. The glycine insertion between S31 and N32 (mutG0, buried between the two exposed paratope regions formed by A29-T30 and I33) was predicted to affect the relative binding contributions of A29 and T30 with opposite and neutralizing effects in terms of ΔΔG (ΔΔG ≈ 1.78 ± 0.87 and ΔΔG ≈ −1.48 ± 0.56). Specifically, T30 was not displaced, and its contribution probably increased because its flexibility along the loop axis (Figure 4b, mutG0) improves hydrophobic contact with Her2 residues P595 and Y554 (Figure 3b). Note that not all of the highlighted residues are active at any given time.

The conformational effects induced by glycine insertions on CDR1 docked to Her2 were predicted by looking at the root mean square fluctuations (RMSF) of the simulated complexes (Figure 4a). Whereas the glycine of mutKG is exposed at the interface with Her2 and determined a slight flexibility increase of both interaction hot spots present in CDR1 (A29-T30 and I33-S34), the glycine insertion in the buried link between the hot spots (mutG0) seems to promote a more consistent CDR1 flexibility, especially in the G28 to T30 region. Cluster analysis also indicates that a different conformational ensemble is explored with respect to A10 wt (Figure 4b), which is consistent with what was previously observed for the unbound ligand. Overall, by assuming the same A10 binding mode and a similar entropic/conformational contribution of CDR1 upon mutation(s), the modeling data suggested that alternative insertions would differently affect the affinity of A10 for Her2, but in both cases, alternative insertions would not drastically modify it, whereas they should provide higher stability in the Her2-bound state.

The docking analysis matched substantially with the experimentally measured affinities. As predicted, the glycine insertion and its position in the CDR1 affected the mutant binding capacity for Her2 differently (Table 1). The affinity of all of the the mutants for Her2 is lower than that of A10 wt (KD = 4.9 nM). A10 mutKG (34.7 nM) performed slightly worse than A10 mutG0 (23.5 nM). Although not suggested by the computational prediction (overall ΔG of A10 mutKG and mutKG.1 are −61.6 ± 10.9 and −55.2 ± 3.8, respectively), the additional mutations present in A10 mutKG.1 had a positive effect on its binding affinity, bringing it closer to that of the wild type (KD = 9.9; Table 1 and Appendix A). Despite the appealing biophysical properties of this latter mutant, it must be remarked that the observed improvements were at the expenses of the nanobody immunogenic potential since the wild type partially humanized sequence was heavily camelized. The final application (in vivo or in vitro) of the reagent(s) should be therefore considered before undertaking a procedure that might increase immunogenicity together with stability.

## 4. Conclusions

Antibody fragments are becoming increasingly popular in both basic research and indrug development thanks to their small dimension, simple engineering, and inexpensive production. While several tools are emerging for the study and design of conventional antibodies, such as AbLIFT, an automated web server that proposes core mutations able to improve contacts between light and heavy chains of a conventional antibody variable domain resulting in variants with a simultaneously improved stability and affinity [38], these tools cannot be applied to single-chain antibodies. Indeed, the stability problems of nanobodies have different reasons requiring dedicated approaches. Our present work showed that the integrated use of different computational tools can be used for a reliable pre-selection of a few candidate mutants to be then tested experimentally. The peculiarity of such mutants is that they present an amino acid insertion and not only point mutations. PROSS was demonstrated to be suitable for the identification of instability hot spots, whereas SeqDesign, PEP-FOLD3, and molecular dynamics-based analyses were useful for evaluating the impact of mutations on both entire nanobodies and short sequences such as the nanobody loops. We opted for a reliable method of ΔΔG calculation to rapidly obtain (few days) the necessary information regarding energy contributions at the residue level. Although the method does not consider entropic and conformational factors (i.e., CDR1 and CDR3 flexibilities are predicted to significantly change in the A10 mutKG variant simulated in absence of Her2), the inferred predictions were highly convergent with experimental data, demonstrating that the approach is accurate enough to be useful from a practical point of view.

The other major innovative result of this work is having demonstrated the relevance of CDRs as potential instability hot spots. Indeed, CDR grafting experiments performed in the past have demonstrated that molecular scaffolds had variable capacity to host heterologous CDRs differing for length and antibody composition in their framework [4,12,39,40]. The consequence is that several constructs are not stable and are eliminated during selection procedures. The new element that has emerged here is that even functional and relatively stable binders isolated from synthetic collections, such as A10, can present instability spots in their randomly generated CDRs. This result should be considered during the design of synthetic libraries because ignoring it might result in the synthesis of several clones, the functionality of which could be strongly reduced and the presence of which can decrease panning efficiency due to clone aggregation and unspecific stickiness.

## Figures and Tables

**Figure 1 biomolecules-11-01362-f001:**
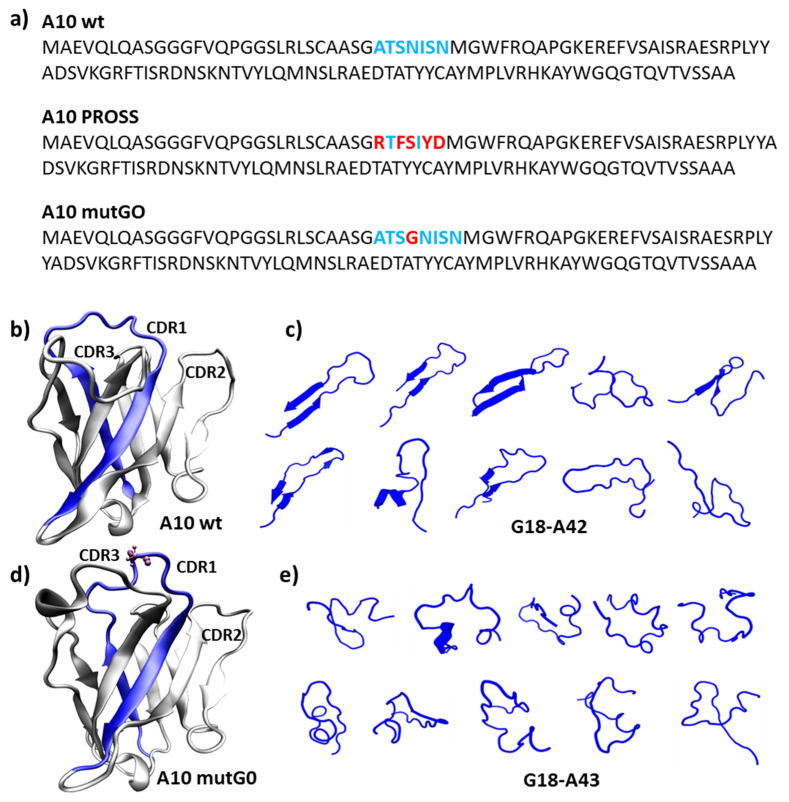
Nanobody sequences and structural effects induced by the insertion of a glycine in A10 wt CDR1. (**a**) Sequences corresponding to A10 wt (CDR1 residues highlighted in blue), to A10 with mutations proposed by PROSS in CDR1 (red), and to the mutant A10 mutG0 (inserted glycine in red); (**b**,**c**) A10 wt and A10 mutG0 initial models (**d**,**e**) and putative three dimensional arrangements of nanobody-derived peptides as predicted by PEP-FOLD3. The initial models have been built by homology. In (**b**–**e**), the peptides (G18-A42 in A10 wt and G18-A43 in A10 mutG0) are highlighted in blue.

**Figure 2 biomolecules-11-01362-f002:**
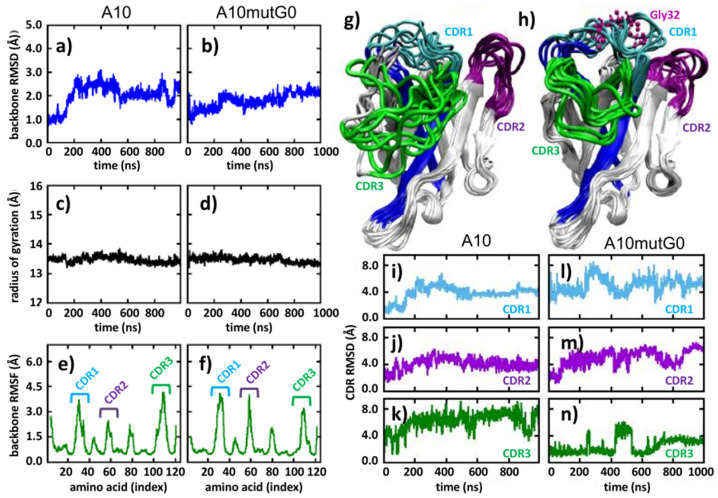
Molecular dynamics analyses. (**a**,**b**) Backbone root mean squared deviation (RMSD), (**c**,**d**) backbone radius of gyration (RMSF), (**e**,**f**) backbone root mean squared fluctuation, (**g**,**h**) overlap of conformations sampled every 100 ns, and (**i**–**n**) CDRs RMSD. (**a**,**c**,**e**,**g**,**i**–**k**) refer to A10 wt, while (**b**,**d**,**f**,**h**,**l**–**n**) refer to A10 mutG0. Color code: CDR1 (cyan), CDR2 (violet), CDR3 (green).

**Figure 3 biomolecules-11-01362-f003:**
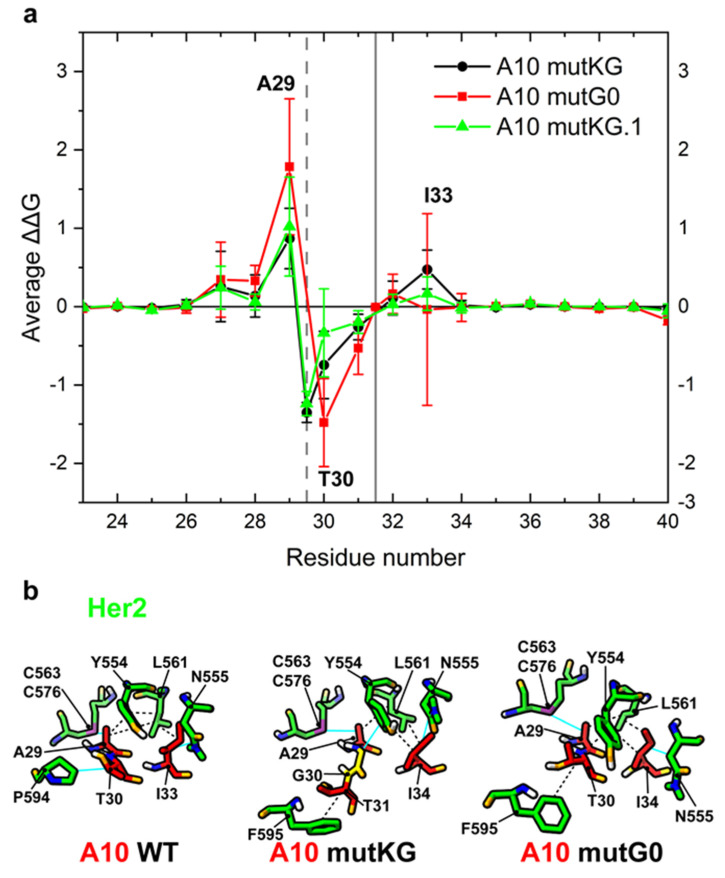
Prediction of per-residue ΔΔG change of A10 variants in the CDR1 region. (**a**) Plot of average and standard deviation of estimated per-residue ΔΔG change between A10 wt and the corresponding A10 variants. The grey vertical bars represent the position of glycine insertion in mutKG and mutKG.1 (dashed line) or mutG0 (solid line). The residues with positive or negative average ΔΔG have reduced or increased predicted binding capacity to Her2, respectively. (**b**) 3D interaction diagram: non-covalent interactions active in at least 50% of simulation frames (see Appendix A for details). For each variant, the representative structure shown is the cluster centroids obtained by cluster analysis of MD simulations. Relevant side chains are highlighted. Hydrophobic interactions are depicted by black dashed lines, Van der Waals interactions by cyan lines. The glycine insertion in mutKG is colored in yellow. Nitrogen atoms are colored in blue, oxygen atoms are colored in orange, and hydrogen atoms are colored in white.

**Figure 4 biomolecules-11-01362-f004:**
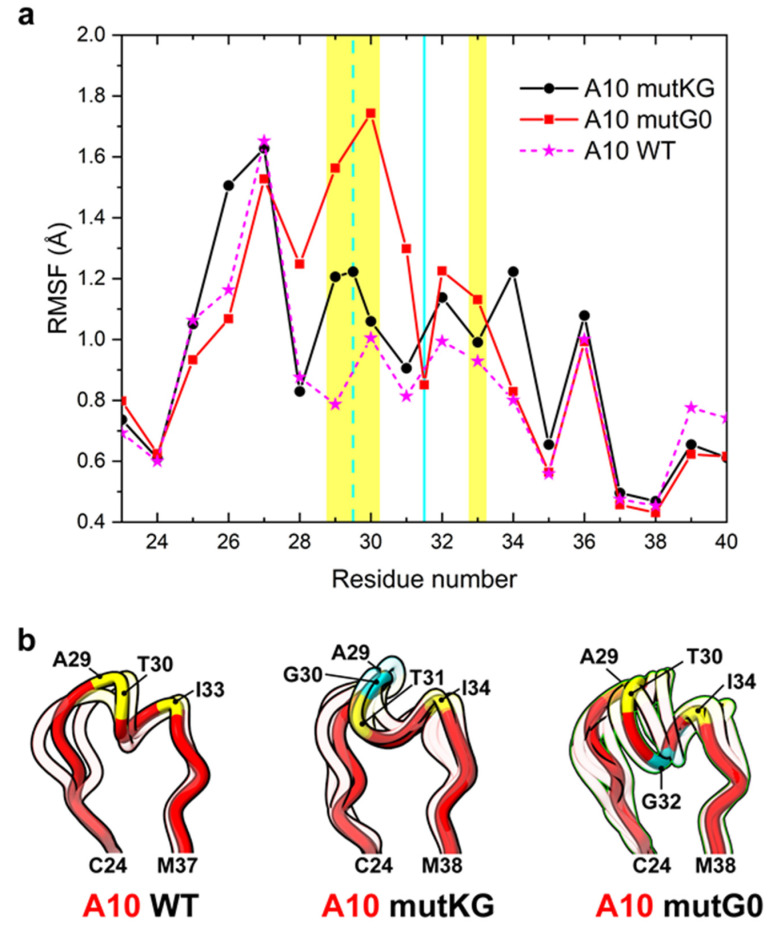
Analysis of CDR1 flexibility of A10 insertional variants interacting with Her2. (**a**) Root mean square fluctuation (RMSF) of CDR1 residues of A10 variants differing for a glycine insertion. The cyan vertical bars represent the positions of glycine insertion in mutKG (dashed line) and mutG0 (solid line). The A10 residues forming the hot spot of interaction with HER2 are highlighted in yellow. (**b**) Centroids from the cluster analysis of MD simulation of each variant are represented in the ribbon style. The centroid of the cluster that includes the higher n° of the simulation frames is opaque, while the centroids of minor clusters are semi-transparent. A10 is colored in red, and the A10 residues forming the hot spot of interaction are colored in yellow; the glycine insertion in mutKG and mutG0 is colored in cyan.

**Table 1 biomolecules-11-01362-t001:** Characteristics of the clones used in the simulations and experiments. CDRs are underlined, point mutations are in red, and insertions in blue.

Variants	Mutations	Amino Acid Sequences	Yields (mg/L)	*K*D (nM)
A10 wt	/	MAEVQLQASGGGFVQPGGSLRLSCAASGATSNISNMGWFRQAPGKEREFVSAISRAESRPLYYADSVKGRFTISRDNSKNTVYLQMNSLRAEDTATYYCAYMPLVRHKAYWGQGTQVTVSSA	5.2	4.9
A10 mutG0	G31a	MAEVQLQASGGGFVQPGGSLRLSCAASGATSGNISNMGWFRQAPGKEREFVSAISRAESRPLYYADSVKGRFTISRDNSKNTVYLQMNSLRAEDTATYYCAYMPLVRHKAYWGQGTQVTVSSA	6.9	23.5
A10 mutKG	G29a	MAEVQLQASGGGFVQPGGSLRLSCAASGAGTSNISNMGWFRQAPGKEREFVSAISRAESRPLYYADSVKGRFTISRDNSKNTVYLQMNSLRAEDTATYYCAYMPLVRHKAYWGQGTQVTVSSA	9.6	34.7
A10 mutKG.1	Q8E, F13L, G29a, S51A, R90K, A91P	MAEVQLQESGGGLVQPGGSLRLSCAASGAGTSNISNMGWFRQAPGKEREFVAAISRAESRPLYYADSVKGRFTISRDNSKNTVYLQMNSLKPEDTATYYCAYMPLVRHKAYWGQGTQVTVSSA	12.8	9.9
A10 mutKG.2	Q8E, F13L, G29a, S51A, R90K, A91P, K109L	MAEVQLQESGGGLVQPGGSLRLSCAASGAGTSNISNMGWFRQAPGKEREFVAAISRAESRPLYYADSVKGRFTISRDNSKNTVYLQMNSLKPEDTATYYCAYMPLVRHLAYWGQGTQVTVSSA	1.2	/

## Data Availability

Data and sequences are all available in either the main text or in the Appendix A.

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
