# Peer review of "CDR1 Composition Can Affect Nanobody Recombinant Expression Yields"

_biomolecules, 2021, doi:10.3390/biom11091362_

Round 1

Reviewer 1 Report

The authors performed computational design of a nanobody and experimentally characterized the designed variants. In the nanobody, they found that mutations in CDR1 can improve the expression level while preserving binding affinity to the antigen. An interesting point in this study is that the authors introduced mutations in a CDR to improve the stability. Antibodies recognize antigens through CDRs, so people usually do not consider mutations in CDRs for stability engineering to preserve the antigen binding. I found this work interesting, and I would recommend the publication in this journal once the authors address a few of my questions and concerns below.

In Figure 1, the authors argued that a set of structured hairpin-like conformations of the WT may represent a potential element of instability and higher flexibility of the Gly mutant may be a reason of better yields. However, it is probably more common that structured proteins would have better stability, in conflict with the authors’ arguments. The better yields were later confirmed by experiments, but the origin might be different from the authors’ arguments. Therefore, I would suggest that the authors discuss this discrepancy. Analysis by PEP-FOLD3 is interesting, but predicting peptide conformational ensemble is not an easy task due to its high flexibility. So I was not sure if PEP-FOLD3 would have been helpful.

In Figure 2, the authors performed MD simulations only one time for each system. This would be problematic because in general trajectories obtained from MD simulations can vary depending on many factors. Therefore, it can be dangerous to draw a conclusion, based only on a single trajectory, even when results are plausible.

In Figure 4b, differences in conformational ensembles among variants is hard to see. May need better visualization/representation.

Author Response

Reviewer 1.

The authors performed computational design of a nanobody and experimentally characterized the designed variants. In the nanobody, they found that mutations in CDR1 can improve the expression level while preserving binding affinity to the antigen. An interesting point in this study is that the authors introduced mutations in a CDR to improve the stability. Antibodies recognize antigens through CDRs, so people usually do not consider mutations in CDRs for stability engineering to preserve the antigen binding. I found this work interesting, and I would recommend the publication in this journal once the authors address a few of my questions and concerns below.

The authors thank the reviewer for the positive general evaluation of their contribution

In Figure 1, the authors argued that a set of structured hairpin-like conformations of the WT may represent a potential element of instability and higher flexibility of the Gly mutant may be a reason of better yields. However, it is probably more common that structured proteins would have better stability, in conflict with the authors’ arguments. The better yields were later confirmed by experiments, but the origin might be different from the authors’ arguments. Therefore, I would suggest that the authors discuss this discrepancy. Analysis by PEP-FOLD3 is interesting, but predicting peptide conformational ensemble is not an easy task due to its high flexibility. So I was not sure if PEP-FOLD3 would have been helpful.

Thank you for having raised the issue concerning flexibility/stability because offers us the possibility to better explain the concept. It is of course true that structured domains are in general more stable than flexible regions but the case of synthetic nanobody CDRs built on fixed scaffolds is a bit particular: they are randomly generated sequences that underwent no evolutionary/functional optimization and they must often adapt to the framework residues to be accommodated in the overall structure. Due to such need, flexibility can provide more stable options than strained conformations. This point has been now better developed in the text.

Concerning the use of PEP-FOLD3, our approach was very pragmatic. We looked for hypotheses on which to work, we got an indication and then moved to experimental evaluation. At least in this specific case, the approach resulted useful. Also this point has been now indicated in the text.

In Figure 2, the authors performed MD simulations only one time for each system. This would be problematic because in general trajectories obtained from MD simulations can vary depending on many factors. Therefore, it can be dangerous to draw a conclusion, based only on a single trajectory, even when results are plausible.

In this particular instance we chose to run a single, 1 microsecond long, simulation instead of running multiple shorter runs. As in the NPT ensemble the system is ergodic, the two approaches are supposed to be equivalent (at least in the limit of an infinite simulation capable of exploring all microstates). We definitively agree with the referee that for building an exhaustive ensemble it would be ideal to randomize the initial system and run multiple long runs from several starting conformations, but being limited by the available computational resources this rigorous approach is not always possible nor necessarily needed for suggesting a possible interpretation of the experimental results. Aware of the referee concern we now underline the raised issue and play-down our conclusions on Figure 2 by reframing its analysis as follows: “Aware of the limits of this analysis, resulting from the data of a single (even if reasonably long) MD run, we believe the enhanced mobility observed in the CDR might explain the fivefold decrease in binding affinity to the target later observed in the experiments.”

In Figure 4b, differences in conformational ensembles among variants is hard to see. May need better visualization/representation.

The Figure 4b has been redesigned to improve its clarity and visualize the effects of mutations on the CDR1 flexibility.

Reviewer 2 Report

In the manuscript entitled: “CDR1 composition can affect nanobody recombinant expression yields” Orlando et al revealed a mutagenesis strategy by inserting an extra glycine into CDR1 sequence to significantly increase the nanobody expression yields. This strategy was based on the identification of CDR1 sequence as an instability hot-spot by the PROSS program and structural information which showed four CDR1 residues directly binding to the antigen. The authors also evaluate the effect of glycine insertion on nanobody conformation and its binding capability to antigen by computational and experimental analysis. Overall, this is an interesting study with a potentially broad interest in protein engineering. However, the manuscript contains several scientific flaws that obviate publishing in the present state.

Major comments:

  1. I can barely read the words from the figures (figure 1, figure 2, figure 3, figure 4). Please update the figures with higher resolution ones. I highly suggest the authors remake all the figures. All the figures have poor quality in this paper. Please also give better explanations for these figures.
  2. Line 84: The authors mentioned crystal structure 6ey0 was used to build the A10 model. Were there any more steps that were used to preprocess your samples? Why did you directly replace the residues instead of building a homology model? Replacing 30% of residues from 6ey0 introduces a large amount of error into your system. Is there any paper from other research groups to support your model-building method? If so, please cite it. Also, it is “residue” instead of “amino acid” in line 84.
  3. Line 107: Please cite the paper for the equation. If the authors derived the equation, please show the derived steps. Please also use the correct mathematical symbols to write the equation. (Please also fix the equation in lines 168-169) It also should be a “log-ratio of likelihoods of individual sequences to predict mutation effects” instead of a “log odds ratio”. The SeqDesign was used as a deep generative model to predict and design the functional sequences. It is using NLP, which is not just simple fitting. If authors want to describe this method, please describe it correctly.
  4. In session 2.4 PEP-FOLD3-based analysis: why did you only submit G18-A42 of A10 and G18-A43 in A10 mutG0 instead of full length?
  5. Based on your paper, you used the A10 wt/Her2 from reference 15. In reference 15, the resolution of Her2 cryo-EM structure is around 4 Å. To discuss the atomistic level structure, resolution ~4 Å structures are not very good to use. I think there are other crystal structures available on the Protein data bank. These high-resolution crystal structures will be better than the EM structures.
  6. From figure 2a and 2b, it shows your two MD system are still equilibrating. The difference in backbone RMSD of A10 is beyond 1 Å. From figure 2a, it shows the system is very dynamic. Please change the unit from nm to Å.
  7. In table 1, the expression yield of A10 mutKG.1 is 12.8 mg/L. Compared with A10 mutKG.1, A10 mutKG.2 has only a further mutation in CDR3 sequence, but the yield of A10 mutKG.2 is only 1.2 mg/L, 10 times less than that of A10 mutKG.1, indicating that CDR3 has a significant effect on the yield of nanobodies. Based on figure 2, CDR3 of A10 mutG0 is intensively arrested. However, the yield of A10 mutG0 is slightly higher than that of A10 wt, meaning that conformation change of CDR3 doesn’t impact much on the yield. Why are these results not consistent? Which of the CDRs dose mostly affect the stability of nanobody?
  8. Figure 4A: the Gly’s RMSF value in the A10 mutG0 mutant is around 0.8 Å. It is lower than residue 31 and 32. What kind of interaction is there between Gly (A10) and Her2? Figure 4B: please remake the figure of figure 4B, please show the interactions between A10 (include mutants) and Her2.

Minor Points:

  1. How did you get the A10 starting model? (homology model or from the experimental structure)
  2. What is your MD system size in session 2.5 molecular dynamics simulations?
  3. Line 143: please fix the references: a) Sondergaard, Chresten R., Mats HM Olsson, Michal Rostkowski, and Jan H. Jensen. "Improved Treatment of Ligands and Coupling Effects in Empirical Calculation and Rationalization of pKa Values." Journal of Chemical Theory and Computation 7, no. 7 (2011): 2284-2295; b) Olsson, Mats HM, Chresten R. Sondergaard, Michal Rostkowski, and Jan H. Jensen. "PROPKA3: consistent treatment of internal and surface residues in empirical pKa predictions." Journal of Chemical Theory and Computation 7, no. 2 (2011): 525-537.
  1. Line 144: why did you use a dodecahedron box as your MD system?
  2. Line 172: please fix the references: a) UCSF ChimeraX: Structure visualization for researchers, educators, and developers. Pettersen EF, Goddard TD, Huang CC, Meng EC, Couch GS, Croll TI, Morris JH, Ferrin TE. Protein Sci. 2021 Jan;30(1):70-82; b)UCSF ChimeraX: Meeting modern challenges in visualization and analysis. Goddard TD, Huang CC, Meng EC, Pettersen EF, Couch GS, Morris JH, Ferrin TE. Protein Sci. 2018 Jan;27(1):14-25.
  1. Please explain why glycine insertion between A29 and T30 has a higher negative effect on the contribution of I33 to the paratope than that of glycine insertion between S31 and N32 in figure 3.

Author Response

Point-to-point reply

Reviewer 2.

In the manuscript entitled: “CDR1 composition can affect nanobody recombinant expression yields” Orlando et al revealed a mutagenesis strategy by inserting an extra glycine into CDR1 sequence to significantly increase the nanobody expression yields. This strategy was based on the identification of CDR1 sequence as an instability hot-spot by the PROSS program and structural information which showed four CDR1 residues directly binding to the antigen. The authors also evaluate the effect of glycine insertion on nanobody conformation and its binding capability to antigen by computational and experimental analysis. Overall, this is an interesting study with a potentially broad interest in protein engineering. However, the manuscript contains several scientific flaws that obviate publishing in the present state.

Thank you for the having acknowledged the potential of our contribution and the suggestions present in your report.

Major comments:

    I can barely read the words from the figures (figure 1, figure 2, figure 3, figure 4). Please update the figures with higher resolution ones. I highly suggest the authors remake all the figures. All the figures have poor quality in this paper. Please also give better explanations for these figures.

Thank you for the indication. Figure graphic and resolution have been improved. Further quality discrepancies due to technical issues (file conversion, data compression) will be hopefully eliminated in collaboration with the Editor

    Line 84: The authors mentioned crystal structure 6ey0 was used to build the A10 model. Were there any more steps that were used to preprocess your samples? Why did you directly replace the residues instead of building a homology model? Replacing 30% of residues from 6ey0 introduces a large amount of error into your system. Is there any paper from other research groups to support your model-building method? If so, please cite it. Also, it is “residue” instead of “amino acid” in line 84.

Thanks a lot for having identified the point. The model was indeed created by homology modelling and the authors apologize for the mistake due to a material error in accepting/refusing modifications during the final text revision

    Line 107: Please cite the paper for the equation. If the authors derived the equation, please show the derived steps. Please also use the correct mathematical symbols to write the equation. (Please also fix the equation in lines 168-169) It also should be a “log-ratio of likelihoods of individual sequences to predict mutation effects” instead of a “log odds ratio”. The SeqDesign was used as a deep generative model to predict and design the functional sequences. It is using NLP, which is not just simple fitting. If authors want to describe this method, please describe it correctly.

This part, that describes the use of SeqDesign, was rewritten to be more precise, according to the description provided by the authors of the paper describing this method. Other two references were added for the use of the log-ratio of likelihoods as a proxy of relative sequence fitness to predict mutation effects.

    In session 2.4 PEP-FOLD3-based analysis: why did you only submit G18-A42 of A10 and G18-A43 in A10 mutG0 instead of full length?

This algorithm has been developed to analyze short peptide dynamic folding and therefore can be applied only to short sequences (5 to 50 residues), such as a CDR and its close flanking regions. However, in this case the analysis of the CDR1 sequence was sufficient since the instability residues indicated by PROSS were all confined in this short sequence.

    Based on your paper, you used the A10 wt/Her2 from reference 15. In reference 15, the resolution of Her2 cryo-EM structure is around 4 Å. To discuss the atomistic level structure, resolution ~4 Å structures are not very good to use. I think there are other crystal structures available on the Protein data bank. These high-resolution crystal structures will be better than the EM structures.

The Cryo-EM structure of Her2 used in reference 15 to study the interaction mode between Her2 and A10 (PDB ID: 6OGE) reported a longer portion of the linker region, bound to Trastuzumab, up to Ala 642. As collected evidence in reference 15 suggests that A10 compete with Trastuzumab in binding to Her2, it was considered the best model to obtain the initial experiment-driven docking solution, which confirmed that a solution compatible with distance restraints could be found to bind in a region partially overlapping with Trastuzumab. Another Her2 model have a solved linker region by x-ray with a slightly higher resolution of 2.92 Å (PDB ID: 6J71). As the two models essentially differ for >1 Å in their atomic coordinates only in the linker region from residue 530 on, it was preferred 6OGE for preparing the system because this flexible region is involved in an interaction and assumed to be in a better conformation with binding residues exposed.

    From figure 2a and 2b, it shows your two MD system are still equilibrating. The difference in backbone RMSD of A10 is beyond 1 Å. From figure 2a, it shows the system is very dynamic. Please change the unit from nm to Å.

Indeed the systems did not reach equilibrium along the simulated time, that we recall is 1 microsend long, but explore a number of states. This is observed more often for A10 (Figure 2a). We thus now include the referee observation when discussing Figure 2a “[…] the wild type protein emerges as a very dynamic system […]”.We have also replaced the units of lengths both in Figure 2 and, for consistency, in Figure 4 and throughout the text.

    In table 1, the expression yield of A10 mutKG.1 is 12.8 mg/L. Compared with A10 mutKG.1, A10 mutKG.2 has only a further mutation in CDR3 sequence, but the yield of A10 mutKG.2 is only 1.2 mg/L, 10 times less than that of A10 mutKG.1, indicating that CDR3 has a significant effect on the yield of nanobodies. Based on figure 2, CDR3 of A10 mutG0 is intensively arrested. However, the yield of A10 mutG0 is slightly higher than that of A10 wt, meaning that conformation change of CDR3 doesn’t impact much on the yield. Why are these results not consistent? Which of the CDRs dose mostly affect the stability of nanobody?

We apologize, the description of the experimental conditions were not precise enough. There are two different levels of instability, namely CDR1 is unstable in the wt nanobody, CDR3 becomes instable only after mutation. Therefore, CDR3 does not contribute to the instability of the original structure, whereas a single point mutation (hydrophilic/hydrophobic substitution) in its sequence strongly affects it. The opposite is for the CDR1 that is initially unstable and can be stabilized by the gly insertion.

    Figure 4A: the Gly’s RMSF value in the A10 mutG0 mutant is around 0.8 Å. It is lower than residue 31 and 32. What kind of interaction is there between Gly (A10) and Her2? Figure 4B: please remake the figure of figure 4B, please show the interactions between A10 (include mutants) and Her2.

The main interactions between the CDR1 of A10 variants and Her2 are now shown in Fig. 3b. Figure 4A has also been improved and a brief description of the residue interactions was added in the text.  

Minor Points:

    How did you get the A10 starting model? (homology model or from the experimental structure)

The point has been addressed (see reply in Major Points)

    What is your MD system size in session 2.5 molecular dynamics simulations?

26289 atoms for A10 and 26255 for A10mutG. We have now added this information in section 2.5

    Line 143: please fix the references: a) Sondergaard, Chresten R., Mats HM Olsson, Michal Rostkowski, and Jan H. Jensen. "Improved Treatment of Ligands and Coupling Effects in Empirical Calculation and Rationalization of pKa Values." Journal of Chemical Theory and Computation 7, no. 7 (2011): 2284-2295; b) Olsson, Mats HM, Chresten R. Sondergaard, Michal Rostkowski, and Jan H. Jensen. "PROPKA3: consistent treatment of internal and surface residues in empirical pKa predictions." Journal of Chemical Theory and Computation 7, no. 2 (2011): 525-537.

The references have been fixed

    Line 144: why did you use a dodecahedron box as your MD system?

A dodecahedron box was used to minimize the n° of water molecules to be added in the system and improve the efficiency of simulations, as the solvated system of the complex contains > 110,000 atoms. The atom distance across periodic boundaries limits of the box is always higher to the short range interaction cut-offs distance (1.0 nm).

    Line 172: please fix the references: a) UCSF ChimeraX: Structure visualization for researchers, educators, and developers. Pettersen EF, Goddard TD, Huang CC, Meng EC, Couch GS, Croll TI, Morris JH, Ferrin TE. Protein Sci. 2021 Jan;30(1):70-82; b)UCSF ChimeraX: Meeting modern challenges in visualization and analysis. Goddard TD, Huang CC, Meng EC, Pettersen EF, Couch GS, Morris JH, Ferrin TE. Protein Sci. 2018 Jan;27(1):14-25.

The references have been fixed

    Please explain why glycine insertion between A29 and T30 has a higher negative effect on the contribution of I33 to the paratope than that of glycine insertion between S31 and N32 in figure 3.

The explanation of this experimental observation has been reported in the text.

Round 2

Reviewer 2 Report

Review: CDR1 composition can affect nanobody recombinant expression yields

(Round 2)

  1. Based on your paper, you used the A10 wt/Her2 from reference 15. In reference 15, the resolution of Her2 cryo-EM structure is around 4 Å. To discuss the atomistic level structure, resolution ~4 Å structures are not very good to use. I think there are other crystal structures available on the Protein data bank. These high- resolution crystal structures will be better than the EM structures.

The Cryo-EM structure of Her2 used in reference 15 to study the interaction mode between Her2 and A10 (PDB ID: 6OGE) reported a longer portion of the linker region, bound to Trastuzumab, up to Ala 642. As collected evidence in reference 15 suggests that A10 compete with Trastuzumab in binding to Her2, it was considered the best model to obtain the initial experiment-driven docking solution, which confirmed that a solution compatible with distance restraints could be found to bind in a region partially overlapping with Trastuzumab. Another Her2 model have a solved linker region by x-ray with a slightly higher resolution of 2.92 Å (PDB ID: 6J71). As the two models essentially differ for >1 Å in their atomic coordinates only in the linker region from residue 530 on, it was preferred 6OGE for preparing the system because this flexible region is involved in an interaction and assumed to be in a better conformation with binding residues exposed.

6OGE’s resolution is 4.36Å. 6J71’s resolution is 2.92Å. There is a difference of 1.44 Å.  This is a big difference. Specifically, you are working on the atomic level in your paper. Beyond 4Å, the structural information you can ascertain is just an outline of the protein. There are also crystal structures, 3MZW with resolution is 2.90 Å, and 3BE1 with resolution is 2.90 Å. Both structures’ lengths are 23-646.

  1. Figure 4A: the Gly’s RMSF value in the A10 mutG0 mutant is around 0.8 Å. It is lower than residue 31 and 32. What kind of interaction is there between Gly (A10) and Her2? Figure 4B: please remake the figure of figure 4B, please show the interactions between A10 (include mutants) and Her2.

The main interactions between the CDR1 of A10 variants and Her2 are now shown in Fig. 3b. Figure 4A has also been improved and a brief description of the residue interactions was added in the text.

Can authors show the interaction diagrams? The type of interactions specifically. For example, is it hydrogen bonds, electrostatic interactions, or hydrophobic effects? From figure 3b, and 4b, I can’t see the interactions. Please make more figures showing the interactions either in 2D or 3D. There are ligplot+, maestro, PoseView and so on. These software can show the interactions. Ligplot+ only can analyze hydrogen bonds and hydrophobic effects. I think chimera also can show the interaction diagrams in 3D. Also, authors can show the zoomed-in view of the binding domain in 3D.

In the figure 3B and 4b, please also label the residues in A10 mutKG and A10 mutG0 like A10 WT.

For the future study:

  • Since your system is very dynamic, you may concern about Makov State Models.

Author Response

6OGE’s resolution is 4.36Å. 6J71’s resolution is 2.92Å. There is a difference of 1.44 Å.  This is a big difference. Specifically, you are working on the atomic level in your paper. Beyond 4Å, the structural information you can ascertain is just an outline of the protein. There are also crystal structures, 3MZW with resolution is 2.90 Å, and 3BE1 with resolution is 2.90 Å. Both structures’ lengths are 23-646.

Despite Her2 models with PDB codes 3MZW and 3BE1 have a resolution of 2.90 Å, they are not adequate for this study because both lack the Her2 domain predicted to interact with A10, in particular the regions 581-590 and above the residue 608 (3BE1), and the region above the residue 579 (3MZW). The region/domain of interaction between Her2 and A10 is now visualized in the new Figure S2. The model with PDB code 6J71 would have been adequate because the 608-642 region is solved (as it is the case of the Cryo-EM model) and the resolution is high (2.92 Å). However, only 6OGE contained the solved 608-646 region at the time in which the work reported in reference 15 was performed. Despite the 6OGE resolution is lower than that of 6J71, it was refined and deposited at atom level resolution and not as a protein outline. Moreover, the docking/MD results of the Her2/A10 complex obtained by using this model were compatible with experimental data (reference 15). Consequently, it was considered the best model to study the interactions between Her2 and A10 variants.

Can authors show the interaction diagrams? The type of interactions specifically. For example, is it hydrogen bonds, electrostatic interactions, or hydrophobic effects? From figure 3b, and 4b, I can’t see the interactions. Please make more figures showing the interactions either in 2D or 3D. There are ligplot+, maestro, PoseView and so on. These software can show the interactions. Ligplot+ only can analyze hydrogen bonds and hydrophobic effects. I think chimera also can show the interaction diagrams in 3D. Also, authors can show the zoomed-in view of the binding domain in 3D.

A new Supplementary Figure 3 was introduced to visualize the region of Her2 which binds to A10 in 3D. As the structures shown in Fig. 3b are centroids of MD simulations, a new table (Supplementary Table S2) was provided with the list of the pairwise interactions and the corresponding type of interaction over the course of at least 10% of the simulation frames. These interactions are now visualized in Fig. 3b as a 3D interaction diagram. The numbering of Her2 residues was updated to follow the sequence deposited in Uniprot (ID P04626), instead of the deposited 3D structure.

In the figure 3B and 4b, please also label the residues in A10 mutKG and A10 mutG0 like A10 WT.

Done

For the future study:

    Since your system is very dynamic, you may concern about Makov State Models.

The authors thank the reviewer for this suggestion and will assess the option in future studies